



# Cosmic noise absorption signature of particle precipitation during ICME sheaths and ejecta

Emilia Kilpua[1], Liisa Juusola[2], Maxime Grandin[1], Antti Kero[3], Stepan Dubyagin[2], Noora Partamies[4,5], Adnane Osmane[1], Harriet George[1], Milla Kalliokoski[1], Tero Raita[3], Timo Asikainen[6], and Minna Palmroth[1,2]

[1]Department of Physics, University of Helsinki, Helsinki, Finland
[2]Finnish Meteorological Institute, Helsinki, Finland
[3]Sodankylä Geophysical Observatory, University of Oulu, Sodankylä, Finland
[4]Department of Arctic Geophysics, The University Centre in Svalbard, Longyearbyen, Norway
[5]Birkeland Centre for Space Science, Bergen, Norway
[6]ReSoLVE Centre of Excellence, Space Climate Research Unit, University of Oulu, Oulu, Finland

*Correspondence to:* Emilia Kilpua (emilia.kilpua@helsinki.fi)

**Abstract.**

We study here energetic ($E > 30$ keV) electron precipitation using cosmic noise absorption (CNA) during the sheath and ejecta structures of 61 interplanetary coronal mass ejections (ICMEs) observed in the near-Earth solar wind between 1997 and 2012. The data comes from the Finnish riometer chain from stations extending from auroral (IVA, 65.2 geomagnetic latitude, 5 MLAT) to subauroral (JYV, 59.0 MLAT) latitudes. We find that sheaths and ejecta lead frequently to enhanced CNA ($> 0.5$ dB) both at auroral and subauroral latitudes, although the CNA magnitudes stay relatively low (medians around 1 dB). Due to their longer duration, ejecta typically lead to more sustained enhanced CNA periods (on average 6–7 hours), but the sheaths and ejecta were found to be equally effective in inducing enhanced CNA when relative occurrence frequency and CNA magnitude were considered. Only at the lowest MLAT station JYV ejecta were more effective in causing enhanced CNA. Some clear 10 magnetic local time (MLT) trends and differences between the ejecta and sheath were found. The occurrence frequency and magnitude of CNA activity was lowest close to midnight, while it peaked for the sheaths in the morning and afternoon/evening sectors and for the ejecta in the morning and noon sectors. These differences may reflect differences in typical MLT distributions of wave modes that precipitate substorm-injected and trapped radiation belt electrons during the sheath and ejecta. Our study also emphasizes the importance of substorms and magnetospheric ULF waves for enhanced CNA.

15 **Keywords.** Magnetospheric physics (Solar wind-magnetosphere interactions), Space plasma physics, ionospheric physics, space weather

## 1   Introduction

Precipitation of high-energy ($E > 30$ keV) electrons from the inner magnetosphere into the Earth's ionosphere is an interesting fundamental plasma process that can have significant consequences to the atmospheric chemistry (e.g., Rodger et al., 2010; 20 Andersson et al., 2014; Seppälä et al., 2015; Turunen et al., 2016; Newnham et al., 2018). This process is also of particular in-





terest for climate models (e.g., Matthes et al., 2017), but its details and external factors governing the efficiency of precipitation are currently not well understood. The electron precipitation can result from substorm injections from the nightside plasma sheet or from the scattering of trapped electrons in the Van Allen radiation belts surrounding the Earth. Precipitation can occur also during pulsating aurora and exhibit modulation at similar time scales as the auroral emission (e.g., Grandin et al., 2017b).

Substorm injections occur predominantly from pre-midnight to midnight (e.g., Aminaei et al., 2006; Beharrell et al., 2015) from geostatationary orbit to about 9 Earth radii (e.g., Spanswick et al., 2010) and electrons then drift around the Earth along the morning side. For both the freshly injected substorm-electrons and the electrons trapped in the Van Allen belts, the precipitation is considered to be primarily facilitated through pitch-angle scattering following interaction with Very Low Frequency (VLF) waves – in particular, due to cyclotron resonance with the lower-band chorus whistler waves. The precipitation via whistler

waves is considered most efficient for a few tens of keV energy electrons, while they mostly give energy to higher energy ($>$ 500 keV) electrons (e.g., Bortnik and Thorne, 2007; Lam et al., 2010). High-amplitude chorus can however cause microburst precipitation of MeV trapped electrons (e.g., Thorne et al., 2005; Osmane et al., 2016; Douma et al., 2019). ULF Pc5 waves also play an important role. The Coroniti and Kennel (1970) theory predicts that ULF waves may enhance energetic electron precipitation by periodically increasing the growth rate of whistler waves. Observational evidence supporting this theory was

provided by Motoba et al. (2013) during a conjunction between the Cluster satellites and a ground-based riometer station.

    Another wave mode that is invoked to precipitate electrons through gyroresonance are electromagnetic ion cyclotron (EMIC) waves. They have been shown to be able to precipitate both MeV (e.g., Usanova et al., 2014) and sub-Mev electrons (e.g., Blum et al., 2019; Hendry et al., 2019), although the efficiency of sub-MeV electron precipitation is still debated. Both chorus and EMIC waves occur outside the high-density plasmasphere, whose outer boundary, the plasmapause, varies significantly in

location with geomagnetic activity and also with the Magnetic Local Time (MLT) (O'Brien and Moldwin, 2003). Inside the plasmasphere, precipitation occurs primarily due to plasmaspheric hiss (Li et al., 2015). Hiss waves can scatter electrons over a wide range of energies, but the scattering times increase significantly with increasing electron energy and decreasing hiss wave power (e.g., Selesnick et al., 2003; Summers et al., 2008). For a few tens to a few hundred keV energies scattering typically occurs from hours to days.

The occurrence of above-describe wave modes has a strong dependence on MLT: Chorus waves are observed predominantly in the morning sector and noon (e.g., Aryan et al., 2014; Li et al., 2009) as they arise from gyroresonance instability due to anisotropic distribution of substorm-injected electrons (Smith et al., 1996). Hiss waves are concentrated on the dayside magnetosphere and extend there to dusk and dawn sectors (e.g., Malaspina et al., 2016) with the strongest amplitude hiss recorded at dusk (Kim and Shprits, 2019). They are believed to result from nonlinear growth of chorus waves when they transfer

into the plasmasphere (e.g., Bortnik et al., 2008; Hartley et al., 2018). The EMIC waves in turn occur predominantly in the duskside of the magnetosphere close to the plasmapause as they are generated by anisotropic ring current proton distributions (Zhang et al., 2016; Saikin et al., 2016).

    Several statistical studies have highlighted differences in geomagnetic and radiation belt responses and in precipitation signatures during different large-scale solar wind drivers (e.g., Huttunen et al., 2002; Borovsky and Denton, 2006; Potapov, 2013;

Kilpua et al., 2015; Asikainen and Ruopsa, 2016; Kilpua et al., 2017b; Shen et al., 2017; Grandin et al., 2017a; Benacquista





et al., 2018), namely interplanetary coronal mass ejections (ICMEs; e.g., Kilpua et al., 2017a) and slow-fast stream interaction regions (SIRs; e.g., Richardson, 2018). ICMEs are generally known to drive the strongest geomagnetic storms while SIRs and following fast streams cause more moderate, but sustained geomagnetic activity (e.g., Borovsky and Denton, 2006; Grandin et al., 2019). At the geostationary orbit ($L = \sim 6.6$), SIR/fast-stream driven storms enhance more efficiently MeV radiation belt electrons than ICME-driven storms that cause deeper and longer depletions. Shen et al. (2017) however showed that at the heart of the outer belt ($L = \sim 4-5$) and low L-shells ICMEs enhance more effectively both MeV and lower-energy electron fluxes. A comparison of energetic particle precipitation during ICMEs and SIRs shows that the former tend to produce higher CNA during the first hours of the geomagnetic storm (Longden et al., 2008), whereas the latter may lead to energetic electron precipitation for up to four days following the storm onset (Grandin et al., 2017a). A case study of an ICME event by Longden et al. (2007) found that most of the CNA observed during the related geomagnetic storm was produced by substorm-injected particles on the nightside. On the dayside, CNA absorption spikes were observed in good correlation with solar wind dynamic pressure pulses.

Most of the studies consider ICMEs as entity, although they consist of two highly different solar wind structures, the turbulent and high dynamic pressure sheath and the considerably smoother ejecta where solar wind dynamic pressure tends to be low (e.g., Kilpua et al., 2017a). As a consequence, the sheath and ejecta lead to very distinct forcing of the magnetosphere and hence very different geomagnetic disturbances (e.g., Huttunen et al., 2002; Nikolaeva et al., 2011; Krauss et al., 2015; Kilpua et al., 2017b): Sheaths perturb more efficiently the high-latitude magnetosphere, while ejecta drive more efficiently strong ring current disturbances. The work by Kilpua et al. (2013) also showed that the interplanetary magnetic field (IMF) and solar wind dynamic pressure fluctuation power in the Pc5 range is significantly higher in the sheath than in the ejecta. As a consequence, it is expected that the inner magnetosphere responds differently to sheath and ejecta. Recent studies (e.g., Kilpua et al., 2015; Lugaz et al., 2015; Turner et al., 2019; Kilpua et al., 2019b) have indeed highlighted that sheaths and ejecta distinct responses in energetic electron fluxes in the radiation belts. A key characteristic of sheaths is that they cause long and sustained depletions of electron fluxes at wide range of energies and $L$-shells. While ejecta often depletes the belt as well, they induce less strong depletions and fluxes also tend to rise during them. The effectiveness of sheaths to deplete the belts is partly related to their strong dynamic pressure that can effectively enhance magnetopause shadowing. On the other hand, the average ULF Pc5 and EMIC wave powers in the inner magnetosphere are higher during sheaths than during ejecta (Kalliokoski et al., 2019; Kilpua et al., 2019b).

In this paper, we perform detailed statistical analysis of energetic electron precipitation during the sheath and ejecta parts of ICMEs. Our key data come from the Finnish riometer chain operated and maintained by Sodankylä Geophysical Observatory (SGO). The precipitation response during sheath and ejecta is studied as well as the dependence on MLT. In particular our investigation of MLT dependence in the coverage of enhanced precipitation and magnitude of precipitation signal in riometer data reveals some interesting features and distinct differences between the sheath and ejecta. We also perform a case study for an ICME event that consisted of particularly turbulent sheath and a flux-rope-type ejecta featuring extended intervals of steadily southward interplanetary magnetic field. This more detailed information on general characteristics of precipitation in ICME substructres is important for understanding the precipitation and physical mechanisms leading to precipitation. The





paper is organized as follows: In Section 2 we present the used data sets. In Section 3 we present our case study of an ICME on July 14–17, 2012 and in Section 4 the statistical results. In Section 5 discuss and summarize our results.

## 2 Research data

Riometers (relative ionospheric opacity metres; e.g., Shain, 1951; Hargreaves, 1969) are passive radars recording the power of

cosmic radio noise that reaches the ground. Their main application is the study of energetic ($E > 30$ keV) particle precipitation, which enhances the ionization in the $D$ region of the ionosphere and leads to absorption of the cosmic radio noise. By subtracting the power measured by a riometer during a geomagnetic event from a quiet-day curve, one obtains the so-called cosmic noise absorption (CNA), which is proportional to the $D$-region electron density enhancement due to energetic particle precipitation. In this study, we use CNA data from five stations of the SGO riometer chain in Finland, whose locations are

indicated with red triangles in Fig. 1 and whose coordinates and observed frequencies are given in Table 1. The stations are equipped with narrow-band wide beam ($60°$) riometers that operate at about 30 MHz frequencies. More details on the CNA derivation (in particular the determination of the quiet-day curve) from SGO riometer observations can be found in sect. 2.1.3 of Grandin (2017).

We removed manually the cases where clear disturbances in the data occurred, e.g., due to interference from human-made

systems. Following the study by Grandin et al. (2017b), we set the CNA threshold value for significant energetic particle precipitation to 0.5 dB, which is clearly above the noise level in the original 1-minute CNA data ($\sim$0.1–0.2 dB).

We have used 10 s International Monitor for Auroral Geomagnetic Effects (IMAGE, https://space.fmi.fi/image/) magnetometer data to derive latitude-longitude maps of the ionospheric equivalent current density with the 2-D Spherical Elementary Current System (SECS) method (Amm, 1997; Amm and Viljanen, 1999; Pulkkinen et al., 2003) as described by Juusola et al.

(2016). Before applications of the 2-D SECS method, a baseline was subtracted from the measurements according to van de Kamp (2013). Figure 1 shows the locations of the magnetometer stations with data available for this event, and an example of the derived equivalent current density distribution (arrows) for one time instance. The background colour shows the curl of the equivalent current density as a proxy for the field-aligned current density. Positive values indicate downward direction of the current and negative values upward direction. Although the curl does not provide correct values for the field-aligned

current density, Weygand and Wing (2016) have shown that it can be used to estimate the location of the region-1/region-2 current boundary. The analysis has been carried out in geographic coordinates, but the magnetic (quasi-dipole, Richmond, 1995) latitude and magnetic local time (MLT) grid is indicated in the plot, for reference. This example shows a strong westward electrojet with downward region-1 current on the poleward flank and upward region-2 current on the equatorward flank, consistent with the early morning sector magnetic local time. The black vertical line in the plot marks the longitude $26°$, along which the

riometer stations are roughly located. We have extracted latitude profiles of the east-west component of the equivalent current density and the curl along this longitude from a time series of such plots and used them to construct Fig. 3.

We also use near-Earth solar wind measurements and geomagnetic indices AL (Davis and Sugiura, 1966) and SYM–H (Wanliss and Showalter, 2006). These data come from the 1-minute OMNI database (King and Papitashvili, 2005) that is





**Table 1.** Summary of the riometers used in this study: The station name and abbreviation, geographic latitude and longitude (CGlat and CGlon), geomagnetic latitude and longitude (CGMlat and CGMlon) as computed from the IGRF 2008 model, L-value, MLT and frequency.

| Station | GGlat | GGlong | CGMlat | CGMlon | L-value | UT | Freq |
| --- | --- | --- | --- | --- | --- | --- | --- |
| | [deg] | [deg] | [deg] | [deg] | | | [MHz] |
| Ivalo (IVA) | 68.55N | 27.28E | 65.24N | 108.29E | 5.5 | +2.97 | 30.0 |
| Sodankylä (SOD) | 67.42N | 26.39E | 64.13N | 106.85E | 5.1 | +2.48 | 30.0 |
| Rovaniemi (ROV) | 66.78N | 25.94E | 63.49N, | 106.11E | 4.8 | +2.46 | 32.4 |
| Oulu (OUL) | 65.08N | 25.90E | 61.75N | 105.14E | 4.3 | +2.76 | 30.0 |
| Jyväskylä (JYV) | 62.42N | 25.28E | 59.01N | 103.37E | 3.7 | +2.65 | 32.4 |

collected at the time of this study primarily from the spacecraft located at the Lagrangian point L1 (ACE and Wind). In the OMNI database, the measurements are propagated from L1 to the nose of the terrestrial bow shock.

The events in this study are selected from the list of sheath regions published in Kilpua et al. (2019a) for the years 1997 to 2012 when nearly continuous CNA data was available from the SGO riometer chain. We used the subset of sheaths that were followed by a clear ejecta (for are review of typical ejecta signatures in interplanetary space, see e.g., Zurbuchen and Richardson, 2006; Kilpua et al., 2017a, and references therein). The list is given Table S1 in Supplement.

## 3 Results

### 3.1 Example event

We will first investigate in more detail an event that occurred on July 14–17, 2012. The solar wind conditions, geomagnetic indices and riometer CNA data are shown in Fig. 2. The panels give a) the IMF magnitude, b) IMF components in GSM coordinates, c) solar wind speed, d) solar wind dynamics pressure and e) subsolar magnetopause position from Shue et al. (1998) model. The next panel gives f) geomagnetic indices SYM–H (blue) and AL index (grey). The bottom four panels (g-j) give riometer CNA data from four stations from which the data was available during the event organized from the highest geomagnetic latitude (MLAT) to the lowest MLAT (IVA, SOD, OUL, and JYV). Note that the CNA plots have different scales. The horizontal lines indicate CNA = 0.5 dB. The time is given in MLT corresponding the location of the SGO chain riometer stations, i.e., here MLT ≈ UT + 2.5 hours.

Our example ICME drove a clear interplanetary shock, featured by abrupt and simultaneous jumps in the magnetic field magnitude, solar wind speed and dynamic pressure. The shock is followed by a very turbulent sheath, while in the ejecta the field direction rotates in a coherent manner. This smooth field rotation is a signature of a magnetic flux rope (e.g., Klein and Burlaga, 1982; Kilpua et al., 2017b). Solar wind speed rises to about 700 km/s in the sheath and then declines monotonically through the flux rope, indicating that the flux rope was expanding while moving past Earth. The solar wind dynamic pressure is also high (up to 30 nPa) and variable in the sheath, but low in the ejecta. As a consequence, the subsolar magnetopause is





strongly compressed during the sheath, reaching geostationary orbit, and its position relaxes back towards the nominal position during the flux rope.

An intense isolated substorm occurs in the beginning of the sheath (AL reaches $\sim -1500$ nT) just after the shock where the magnetic field exhibits large-amplitude fluctuations with predominantly southward orientation. At this time, SYM–H however

is positive signalling the Storm Sudden Commencement (SSC). The auroral activity subsides for the rest of the sheath due to fluctuating fields having primarily northward orientation. At the flux rope leading edge, the magnetic field turns strongly southward and rotates then slowly back towards more northward position. The SYM–H index decreases rapidly and strong substorms occur. Although the southward field component weakens as the flux rope progresses, its southward orientation keeps auroral activity and SYM–H at disturbed levels. Geomagnetic activity subsides in the end of the flux rope due to a northward

$B_Z$ period, but enhances again for a few hours during the "back sheath", which forms due to interactions of the ICME with the trailing solar wind. The minimum SYM–H value is $-134$ nT (i.e., intense geomagnetic storm) and it is reached at the front part of the ejecta.

The bottom panels of Fig. 2 show that at the auroral riometer stations IVA and SOD and at the subauroral station OUL enhanced CNA (CNA $> 0.5$ dB) was observed already at the front part of the sheath when the isolated substorm occurred

around 20-24 MLT. Some sporadic enhanced CNA values, but lower in amplitude, were also measured during the northward $B_Z$ part of the sheath at these stations from midnight to morning hours. During this period, solar wind dynamic pressure was high and variable. At the front part of the flux rope when the SYM–H index is most disturbed, the CNA levels hardly reach 0.5 dB from IVA to OUL, but enhanced absorption is recorded at the lowest MLAT station JYV ($\sim 10$–$20$ MLT). From about 22 MLT, on July 15, enhanced absorption takes place at all stations until about 6 MLT (extending at 14 MLT at OUL) on July

16, coinciding with continuous substorm activity in the mid-part of the flux rope. During the end part of the flux rope when $B_Z$ is close to zero/positive, CNA values are mostly below 0.5 dB. The absorption intensifies again during the southward fields and related substorm activity in the "back sheath" from 2 to 14 MLT with magnitudes similar or even higher than observed during the earlier intervals of this event. We note that CNA reaches momentarily levels greater than 2 dB at IVA, greater than 4 dB at SOD, greater than 2.5 dB at OUL and greater than 1.5 dB at JYV.

Next we will compare ionospheric currents and ground Pc5 pulsations and precipitation signatures. Fig. 3a shows the east-west component (positive east) of the ionospheric equivalent current density ($Jeq_y$) as a function of MLT at the geomagnetic longitude of the riometer stations (here MLT $\approx$ UT + 2.5 hours). CNA have also been included in the plot. Fig. 3b shows the curl of the equivalent current density, with red colour indicating downward and blue upward field-aligned current. Pc5 pulsation power, estimated by integrating $Jeq_y$ wavelet power over the Pc5 period range (150–600 s), is shown in Fig. 3c.

Figure 3a generally shows a stronger signature of either positive (red) or negative (blue) values at lower latitudes and a slightly weaker signature of opposite polarity at higher latitudes at a given time. The stronger signature can be interpreted as the eastward or westward auroral ionospheric electrojet and the weaker signature as polar cap currents. The eastward electrojet typically dominates in the local afternoon and the westward electrojet around midnight and in the morning. Occasionally, especially in the pre-midnight sector, both electrojets can exist at the same time, the eastward electrojet located equatorward

of the westward electrojet. Fig. 3a illustrates the equatorward expansion and poleward contraction of the auroral electrojets





during ambient southward and northward IMF conditions, respectively. Shorter time-scale poleward expansions of the poleward electrojet boundary are related to substorm expansion phase activity. According to Fig. 3a and b, there is downward region-1 current and upward region-2 current in the poleward and equatorward parts of the westward electrojet, respectively, and upward region-1 current and downward region-2 current in the poleward and equatorward parts of the eastward electrojet. Apart from

the beginning of the event, when the magnetosphere was being compressed by the increasing dynamic pressure, energetic electron precipitation can generally be observed in the region dominated by region-2 current, more typically upward region-2 current (blue) than downward region-2 current (red). Region-2 current is believed to map to the ring current (e.g., Zheng et al., 2006), indicating that the magnetospheric source of the observed energetic electron precipitation could be in the same area.

Finally, Fig. 3c shows that the power of Pc5 pulsations is enhanced during the times when enhanced precipitation occurs,

in particular in the middle of the flux rope and close to the "back sheath". We also note that Pc5 power is elevated during the dominantly northward IMF in the sheath region when enhanced CNA was also observed.

### 3.2    Statistical results

Our statistical study includes in total 61 sheaths and ejecta. The mean duration of the sheaths in our data set is 9.5 hours with a standard deviation of 4.4 hours. The shortest sheath lasted for 2.5 hours and the longest one for 21.7 hours. The average

duration of the ejecta is 21.5 hours with a standard deviation of 13.3 hours. The shortest ejecta lasted for 4.2 hours and the longest one for 63 hours. Table 2 gives the number of sheaths and ejecta from the 61 events investigated when the CNA data are available. Stations ROV and JYV have several events without the CNA data, while for the other stations there are data for nearly all events.

For each station, Table 2 lists the number of events with enhanced CNA (i.e., CNA > 0.5 dB), the mean time $<t_{CNA > 0.5\ dB}>$

in minutes during the sheath and ejecta when enhanced CNA occurred and the mean relative occurrence of enhanced CNA $<n_{CNA > 0.5\ dB}/n_{all}>$, i.e. the mean value of the ratios of samples with CNA > 0.5 dB divided by all samples within a given sheath and ejecta. The last column gives the mean CNA magnitude <CNA> calculated considering only values > 0.5 dB. Nearly all (> 80%) sheaths and ejecta included in this study resulted in enhanced CNA except at the lowest MLAT station JYV where only 44% of the sheaths and 65% of the ejecta resulted in CNA > 0.5 dB. In addition, only 67% of the sheaths induced

enhanced CNA at IVA, while this was the case for 87% of the ejecta. The ejecta are also associated with longer periods of enhanced CNA than the sheaths. At auroral stations SOD and ROV, sheaths induce enhanced CNA on average for 2.5-3 hours, while the ejecta-related enhanced CNA lasts on average 6-7 hours. JYV exhibits clearly the shortest time of enhanced CNA from all investigated stations both for the sheaths and ejecta, for the sheaths the mean duration being only 40 minutes. Longer periods of enhanced CNA for the ejecta could be at least partly explained by their longer duration (on average twice as long

compared to the sheaths in our study, see above). Sheaths and ejecta have also very different IMF $B_Z$ profiles, ejecta having smooth field changes, being thus capable of providing sustained periods of southward IMF, while sheaths are more turbulent in nature. The mean relative occurrences are more similar between the sheath and ejecta. This suggests that these structures cause almost equally effectively precipitation. The mean magnitude of CNA is also very similar at SOD, ROV and OUL between



**Table 2.** The first two columns give the number of sheaths and ejecta for which CNA data were available and for which enhanced CNA (CNA > 0.5 dB) occurred (the percentage of the events with enhanced CNA from the events for which data were available is shown in parentheses). The next columns give the mean time when enhanced CNA was observed (in minutes), the mean relative occurrence of enhanced CNA (defined as the ratio of samples $n$ with enhanced CNA to all samples), and the mean magnitude of CNA (considering only values > 0.5 dB). The total number of events analysed is 61.

|  | IVA | SOD | ROV | OUL | JYV |
|---|---|---|---|---|---|
| Sheath |  |  |  |  |  |
| Events with data | 58 | 58 | 46 | 56 | 48 |
| Events with enhanced CNA | 39 (67%) | 50 (86%) | 42 (91%) | 46 (82%) | 21 (44%) |
| $<t_{\text{CNA} > 0.5\,\text{dB}}>$ [min] | 110.3 | 172.8 | 157.5 | 124.6 | 40.1 |
| $<n_{\text{CNA} > 0.5\,\text{dB}}/n_{\text{all}}>$ | 0.27 | 0.39 | 0.41 | 0.30 | 0.088 |
| <CNA> [dB] | 1.29 | 1.44 | 1.35 | 1.36 | 1.32 |
| Ejecta |  |  |  |  |  |
| Events with data | 59 | 58 | 47 | 59 | 48 |
| Events with enhanced CNA | 55 (83%) | 44 (95%) | 45 (84%) | 53 (90%) | 31 (65%) |
| $<t_{\text{CNA} > 0.5\,\text{dB}}>$ [min] | 223.5 | 376.6 | 374.2 | 327.5 | 143.5 |
| $<n_{\text{CNA} > 0.5\,\text{dB}}/n_{\text{all}}>$ | 0.28 | 0.40 | 0.41 | 0.31 | 0.12 |
| <CNA> [dB] | 1.09 | 1.40 | 1.37 | 1.30 | 1.47 |

the sheath and ejecta. At IVA in turn the sheaths have larger <CNA>, while at JYV the ejecta induce somewhat stronger CNA values on average.

### 3.2.1 Superposed epoch analysis

The superposed epoch analysis results of solar wind conditions and geomagnetic indices during the studied ICME events are
5  shown in Fig. 4. We use here the double superposed epoch analysis (e.g., Kilpua et al., 2015) where the sheaths and ejecta are all resampled to the same average duration (9.5 hours and 21.5 hours, respectively). The parameters shown in the figure are (4a) interplanetary magnetic field (IMF) magnitude, (4b) IMF north-south magnetic field component ($B_Z$), (4c) 1-minute root-mean-square of the magnetic field ($B_{rms} = \sqrt{<B^2> - <B>^2}$) as the measure of IMF fluctuation level, (4d) solar wind speed, (4e) solar wind dynamic pressure, and (4f) AL and (4g) SYM–H indices. The thick curves in each panel give the
10  median values, and the thinner curves indicate the upper and lower quartiles.

    The sheath and ejecta in our data set have on average similar magnetic field magnitudes and solar wind speeds, both enhanced clearly with respect to the ambient values. They however feature some clear differences; sheaths have much higher level of magnetic field fluctuations and higher solar wind dynamic pressure. The latter is due to generally much higher densities in sheaths than in ejecta (data not shown, see e.g., Kilpua et al. (2017b)). The sheath and ejecta cause quite comparable AL





activity, while SYM–H reaches clearly more disturbed values during the ejecta. We also note that while for the ejecta even the lower quartile SYM–H curve reaches the weak storm level (SYM–H $< -30$ nT), a significant fraction of sheaths have mostly positive SYM–H throughout. This occurs in particular in the front part of the sheath, following the shock impact.

Figure 5 shows the superposed epoch analysis results of 1-minute CNA values measured at the five stations listed in Table
1. Again, the sheath and ejecta are resampled to the same average duration. The lower quartile CNA curves are approximately at the same level throughout the ICME and also only slightly enhanced when compared with the ambient solar wind. They presumably represent the population that had also the weakest geomagnetic response. When the median and upper quartiles are considered, the effect of the ICME is more striking, in particular when considering the upper quartiles curve and the subauroral stations OUL and JYV. The CNA response in the superposed plot is also of comparable magnitude for the sheath and ejecta,
except at JYV where the response is strongest in the ejecta. CNA values tend to be somewhat higher in the middle of the sheath and ejecta than at their boundaries, but no obvious temporal trends can be detected from the superposed plot. This is consistent with the fact that southward fields in sheaths and ejecta (during which the strongest CNA is expected) can occur anywhere in these structures (e.g., Huttunen et al., 2005; Kilpua et al., 2019a). Flux rope-type ejecta show a solar cycle trend in their magnetic polarity (e.g., Bothmer and Schwenn, 1998), but since our data set covers almost two solar cycles no clear effect from
this is expected. For all of the investigated stations, the solar wind trailing the ICME has higher CNA levels than the preceding solar wind. This likely reflects the effect of high-speed stream with Alfvénic fluctuations that often follows the ICME and that keeps also geomagnetic activity elevated (see two bottom panels of Fig. 4).

### 3.2.2 MLT variations

The superposed epoch analysis performed in the previous section gives only a rather limited understanding of the CNA response
that can be very sporadic and have a strong MLT dependence. In this section, we will therefore investigate differences in the CNA response during the sheath and ejecta as a function of MLT.

Figure 6 shows the relative frequency of occurrence of CNA $> 0.5$ dB periods as a function of MLT for the five stations under investigation. Each 1-hour bin shows the fraction of 1-minute data points with CNA $> 0.5$ dB considering all samples measured during the sheaths and ejecta. In Fig. 7 we also give the medians and interquartial ranges (IQRs) calculated over the
relative frequency of occurrence of CNA $> 0.5$ dB of individual events. Note that for several cases the majority of events had zero occurrence of enhanced CNA in a given MLT bin and medians are thus zero. The horizontal lines indicate the bootstrapped standard error of the median calculated using 10 000 random subsets of the original median data.

We will first study the occurrence of enhanced CNA at auroral stations IVA, SOD and ROV. Figure 6 shows that for the sheaths the occurrence peaks in morning hours from 4 to 7 MLT, being $\sim 50 - 60\%$ at SOD and ROV and $\sim 35 - 45\%$ at
IVA. At SOD and ROV frequency is also high ($\sim 40 - 45\%$) in afternoon and evening hours. It stays over 30% for all MLT bins at SOD and for the majority of bins at ROV, while at IVA drops below 30% in a few bins. Figure 7 shows that the largest fraction of sheaths with zero/low frequency of enhanced CNA occur at noon and at IVA also in early morning/late evening hours. The largest fraction of sheaths with high occurrence frequencies occur in morning and afternoon/early evening MLTs (at ROV and SOD), consistent with Fig. 6. In general IQRs are wide suggesting a large spread in occurrence frequencies. For





the ejecta, Fig. 6 shows that the frequency of enhanced CNA at auroral stations peaks in morning and noon hours, at SOD and ROV being $\sim 50 - 60\%$ from 3 to 12 MLT, while at IVA the highest frequencies (about $35 - 40\%$) occur in the morning sector. Similar to sheaths, at SOD and ROV the majority of MLT bins have over 30% frequency, but at IVA it is below 30% from noon to midnight and early morning. Figure 7 reveals again a large variations in the median frequencies and wide IQRs. The

median frequencies have a bi-modal MLT distribution with a strong peak at morning hours that extends to noon and another, considerably weaker peak, at evening hours.

We will next investigate the occurrence frequency of enhanced CNA at subauroral stations OUL and JYV. Figure 6 shows that at OUL enhanced CNA occurs still relatively frequently. The frequency drops below 30% only for a few MLT bins in morning and evening hours for the sheaths and from 18 to 1 MLT for the ejecta. For the sheaths the highest frequencies ($\sim 40 - 50\%$)

occur at 5 and 16 MLT and at noon. For the ejecta the frequency peaks from 10 to 12 MLT ($\sim 50\%$) and is relatively high also in early morning and afternoon hours. At the lowest MLAT station JYV the occurrence frequency drops below 25% for all MLT bins for both the sheaths and ejecta. The enhanced CNA activity is most frequent in noon, early morning and late evening hours and almost vanishes in the late morning and afternoon sectors. Figure 7 reveals for the ejecta at OUL a clear drop in the medians of occurrence frequencies in the afteroon/event sector and at midnight, while a distinct peak occurs at noon. The

upper quartiles are particularly wide for most MLTs. At JYV, the majority of MLTs had zero occurrence of CNA $> 0.5$ dB for both the sheaths and ejecta.

The magnitude of the CNA response is investigated in Fig. 8, that shows the medians and IQRs of the median CNA response for the sheaths and ejecta separately as a function of MLT. Only CNA values $> 0.5$ dB are considered and we have plotted only the cases when the number of events with enhanced CNA exceeded five. The horizontal lines indicate, similar to Fig. 7,

the bootstrapped standard error of the median calculated using 10 000 random subsets of the original median data.

For the sheaths, the largest absorption from IVA to OUL takes place from noon to evening hours, while for the ejecta absorption is strongest from morning sector to noon hours. At JYV the magnitude of CNA is largest at noon for the ejecta, while for the sheaths only small fraction of events reached CNA $> 0.5$ dB values. Figure 8 also shows that in general there are no drastic differences in the magnitude of CNA between the sheaths and ejecta. At IVA and OUL, sheaths have some MLT bins

with considerably larger upper quartiles than the ejecta. Another interesting feature is that the magnitude of CNA is generally weakest near midnight for both the sheath and ejecta.

### 3.2.3   Dependence on driver characteristic

Figure 9 shows the overall properties of the subsets of the sheath and ejecta that were associated with strong and weak absorption. The division to these subsets is made by first calculating median CNA for each event (considering only values $> 0.5$ dB)

and setting the median to zero for those events for which CNA values did not exceed 0.5 dB at all. Then weak (strong) absorption sheath subset is composed of those sheaths whose median CNA is below (above) the lower (upper) quartile for all sheaths and similarly for the ejecta. The figure gives the medians (black bars) and IQR (coloured bars) of the a) the magnetic field magnitude, b) solar wind speed, c) dynamic pressure, d) RMS of the magnetic field, and e) AL, and f) SYM–H indices. The magnetic field magnitude and solar wind speed of the ejecta are calculated using the first six hours as they typically show





a declining trend throughout the event (also visible in our example event shown in Fig. 2). For other parameters, the values are calculated for the whole sheath and ejecta intervals, respectively.

According to Fig. 9, strongest absorption generally occurs during sheaths and ejecta that have strong magnetic fields and large speeds and that are clearly more geoeffective in terms of both AL and SYM–H indices than those related to weakest absorption. The strongest CNA sheaths have also higher solar wind dynamic pressure and IMF fluctuation levels than the sheaths related to the weakest CNA. The dynamic pressure and IMF fluctuations are much weaker in ejecta and thus they do not show such obvious trends.

We note that the variability in the properties of sheath and ejecta that caused the strong CNA is however large, as indicated by generally wide IQRs. It is important to note that slower and weaker sheaths and ejecta that are only mildly geoeffective can also cause significant CNA activity; in particular, this is the case for the sheaths when their geoeffectiveness in determined in terms of SYM–H.

## 4   Summary and Discussion

We have studied in this paper riometer cosmic noise absorption (CNA) during key structures in ICMEs, namely the sheath and ejecta. Our data set consists of 61 sheaths and ejecta that were identified in the near-Earth solar wind during 1997–2012. We performed a superposed epoch analysis by resampling the sheaths and ejecta to the population mean duration and investigated the CNA response as a function of magnetic local time (MLT) and geomagnetic latitude (MLAT). Five stations of the Finnish riometer chain from auroral $\approx 65°$ (IVA) to subauroral $\approx 59°$ (JYV) MLAT were considered in the study. These stations map from $L = 5.5$ to $L = 3.7$, covering roughly the outer parts and the heart of the outer radiation belt. The comparison of precipitation between the sheath and ejecta was motivated by distinct differences in their typical solar wind conditions and geomagnetic responses, summarized in the Introduction and shown also in the superposed epoch analysis results in this paper.

Our study shows that sheaths and ejecta frequently cause enhanced precipitation (defined here as CNA > 0.5 dB). The interval of enhanced CNA lasts on average 2.5–3 hours for the sheaths and 6.5–7 hours for the ejecta, which corresponds to about 40% of their durations (note that ejecta are considerably longer in duration than sheaths, in our data set approximately twice as long). When the relative frequency of enhanced CNA was considered, we found the sheaths and ejecta to be almost equally effective in inducing enhanced CNA. The fraction of ICMEs causing enhanced CNA was high (mostly $\sim 80 - 95\%$) at auroral latitudes (IVA, SOD and ROV) and at the subauroral station OUL, but dropped clearly at the lowest MLAT station JYV. In particular, the fraction of sheaths causing enhanced CNA as well as the average duration and fraction of CNA > 0.5 dB values were considerably lower at JYV than at the other stations. The fraction of sheaths inducing enhanced CNA at the highest MLAT station IVA was also lower (67%) than for the other auroral stations. The fact that ejecta are associated with more enhanced CNA at JYV than sheaths likely results from their stronger geoeffectiveness measured with SYM–H, which implies to an equatorward motion of the auroral oval and of the region of energetic electron precipitation. The auroral activity was in turn at more similar level between the sheath and ejecta, as indicated by the AL index.





The magnitude of enhanced CNA was also found to be roughly at similar levels for the sheath and ejecta for all investigated stations. At IVA and OUL the sheaths had clearly higher upper quartiles of CNA magnitude for a few MLT bins, suggesting that some sheaths induce sporadically relatively strong precipitation. On average, the magnitude of CNA was relatively modest both during the sheath and ejecta, medians being mostly around 1 dB or a bit lower and upper quartiles below 2 dB. We note that

these are similar with the magnitudes for the ULF-associated high-speed stream (HSS) events studied by Grandin et al. (2017a). For the substorm associated events, the authors found median CNA of 1.5–2.4 dB in the morning sector. This is interesting as sheaths and ejecta are on average associated with stronger geomagnetic disturbances, both in terms of SYM–H/Dst and auroral indices than HSSs (e.g., Kilpua et al., 2017b, and references therein). HSSs and preceding slow-fast stream interaction regions (SIRs) cause in turn effectively sustained and moderate level substorm activity that is associated with continuous excitation

of chorus waves in the inner magnetosphere (e.g., Miyoshi et al., 2013; Jaynes et al., 2015). Asikainen and Ruopsa (2016) also found using NOAA/Polar Orbiting Environmental Satellites (POES) energetic electron precipitation data that at $> 30$ keV energies HSSs had a dominant contribution over ICMEs, although annual fluxes were about at the same level. Another difference between the HSS- and ICME-associated events is that for ICMEs, enhanced CNA occurred less frequently and was weaker in magnitude at IVA than at SOD and ROV (also the case for our example event shown in Fig. 2), while for the HSS

events in Grandin et al. (2017a) this was the opposite. This could be related to equatorward expansion of auroral oval during more geoeffective ICMEs. For example, Holappa et al. (2014) showed that ICMEs cause relatively more activity at sub-auroral stations than HSSs.

We found that the sheath and ejecta cause enhanced CNA at all MLTs. Only at the lowest MLAT station JYV did the CNA activity almost vanish in the morning and evening sectors. Our study also revealed some clear MLT dependencies and

differences in the CNA response between the ejecta and sheath. Sheaths caused high occurrence frequencies of enhanced CNA more evenly over a wider range of MLTs than the ejecta with bias also towards afternoon and evening hours. For the ejecta in turn the highest occurrence frequencies occurred at noon and in the morning sector. The magnitude of the enhanced CNA for the sheaths was highest in the afternoon sector and for the ejecta at noon and during early morning hours. For both sheath and ejecta, the occurrence frequencies and CNA magnitudes were relatively small near midnight.

The reasons for these tendencies are not obvious. In the Grandin et al. (2017a) study, ULF-type CNA events peaked strongly on the dayside. The authors concluded that they originate primarily from substorm-injected electrons in the nightside plasma sheet that drift to the dayside in about 30 minutes and precipitate due to wave-particle interactions with chorus waves (see also e.g., Østgaard et al., 1999). As discussed in the Introduction, chorus waves occur from the morning sector to noon which can effectively precipitate injected electrons as well as trapped radiation belt electrons. The excitation of chorus waves is modulated

by ULF waves that are generally enhanced both in the ejecta and in sheaths (Kalliokoski et al., 2019). This thus implies that the magnitude of precipitation from direct substorm injections is lower compared to the magnitude of precipitation associated with wave-particle injections later during the drift around Earth. Another factor explaining low occurrence frequencies and magnitudes close to midnight could be that on the nightside free electrons produce efficiently negative ions, decreasing electron density, while on the dayside they can stay longer due to competing photo-detachment process (e.g., Rodger et al., 2012). Some

sheaths can also induce relatively strong and sustained absorption near noon. The physical mechanism could be related to the





impact of interplanetary shock at the beginning of the sheath and pressure pulses within the sheath leading to increased ULF Pc5 power in particular at the dayside magnetosphere (e.g., Kepko and Spence, 2003; Claudepierre et al., 2008; Wang et al., 2017) and consequently to the enhancement of whistler mode wave growth rates and precipitation (e.g., Belakhovsky et al., 2017).

The strong and relatively frequent enhanced CNA for the sheaths during afternoon hours could result from scattering of trapped radiation belt electrons by EMIC waves when outside the plasmasphere and by hiss when inside the plasmasphere (e.g., Rodger et al., 2007; Clilverd et al., 2008). The latter may occur also at geomagnetically quieter times at lowest MLAT stations when the plasmapause expands to higher L-shells. According to Kalliokoski et al. (2019) and Kilpua et al. (2019b), EMIC waves are particularly enhanced in sheaths, also in those sheaths that are only mildly geoeffective or not geoeffective

at all. As mentioned in the Introduction, it is currently not clear how effectively EMIC waves can precipitate electrons from a few keVs to few hundred keVs which are believed to be the main source of CNA in riometer signal (e.g., Kellerman et al., 2015). In the recovery phase of a storm low intensity chorus waves are also observed at all MLTs (Bingham et al., 2019) that could contribute to precipitation and enhanced CNA, in particularly for the ejecta that have more symmetric ring current than sheaths (e.g., Pulkkinen et al., 2007). The enhanced precipitation during evening hours can result from substorm injections.

This region maps with the location of the westward electrojet (and upward R2 current).

    Our example event highlighted the temporal variability in the occurrence and magnitude of CNA at each investigated station during a particularly turbulent sheath and smooth flux-rope-type ejecta. Enhanced CNA was observed throughout the event when substorms occurred and when the strongest ULF Pc5 power was detected. The latter supports the conclusion made above and in previous studies that ULF waves play a key role in controlling the growth of chorus waves and thus indirectly modulating

the energetic electron precipitation (e.g., Coroniti and Kennel, 1970). The importance of substorms was also highlighted in our statistical study. Those sheaths and ejecta that were related to the strongest CNA were associated with considerably more disturbed AL than those cases that caused the weakest CNA response. The SYM–H index in turn does not need to be strongly disturbed. Sheaths in particular can cause significant CNA activity even though SYM–H does not even reach the weak storm levels ($-30$ nT). For our example event, enhanced CNA occurred at auroral latitudes also during the the parts of the sheaths

with northward IMF when substorm activity subsided. This CNA activity is likely related to high and variable solar wind dynamic pressure in the sheath. Our results are therefore also consistent with Longden et al. (2008) who related nightside CNA to substorms and dayside CNA to solar wind dynamic pressure pulses during an ICME event (see the Introduction). We further found, as expected, that strongest precipitation occurs during fast and strong (in terms of magnetic field) solar wind drivers. The sheaths associated with strongest CNA had also tendency towards large solar wind dynamic pressure and magnetic field

fluctuation levels.

    To conclude, the sheaths and ejecta can both effectively induce moderate-level CNA activity over a wide range of MLTs. These ICME structures appear however less effective than high-speed streams in inducing strong CNA. The found MLT trends in the magnitude and occurrence frequencies of enhanced CNA ($> 0.5$ dB) between the sheath and ejecta can reflect differences in the dominant occurrence of wave modes that precipitate the substorm-injected and radiation belt electrons. It will be left for



a future study to investigate in more detail the MLT distributions and wave power for waves, in particular chorus waves, in the inner magnetosphere during the sheath and ejecta.

*Competing interests.* The authors declare that they have no conflict of interest.

*Acknowledgements.* The Finnish Centre of Excellence in Research of Sustainable Space, funded through the Academy of Finland grant number 312351. We also gratefully acknowledge the Academy of Finland (grant numbers ). E. Kilpua acknowledges Academy of Finland project 1310445. This project has received funding from the European Research Council (ERC) under the European Union's Horizon 2020 research and innovation programme (ERC-COG 724391). The work by NP is partly supported by the Norwegian Research Council under CoE contract 223252 and contract 287427. T.A. acknowledges support by the Academy of Finland to the ReSoLVE Center of Excellence (project No. 307411) and to the PROSPECT (project no. 321440) research project.

We thank the institutes who maintain the IMAGE Magnetometer Array: Tromsø Geophysical Observatory of UiT the Arctic University of Norway (Norway), Finnish Meteorological Institute (Finland), Institute of Geophysics Polish Academy of Sciences (Poland), GFZ German Research Centre for Geosciences (Germany), Geological Survey of Sweden (Sweden), Swedish Institute of Space Physics (Sweden), Sodankylä Geophysical Observatory of the University of Oulu (Finland), and Polar Geophysical Institute (Russia). Sodankylä Geophysical Observatory is acknowledged for the riometer data. The authors are thankful to all of the Van Allen Probes, Wind, and OMNI teams for making their data available to the public. The OMNI data were obtained through CDAWeb (https://cdaweb.sci.gsfc.nasa.gov/index.html/).





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

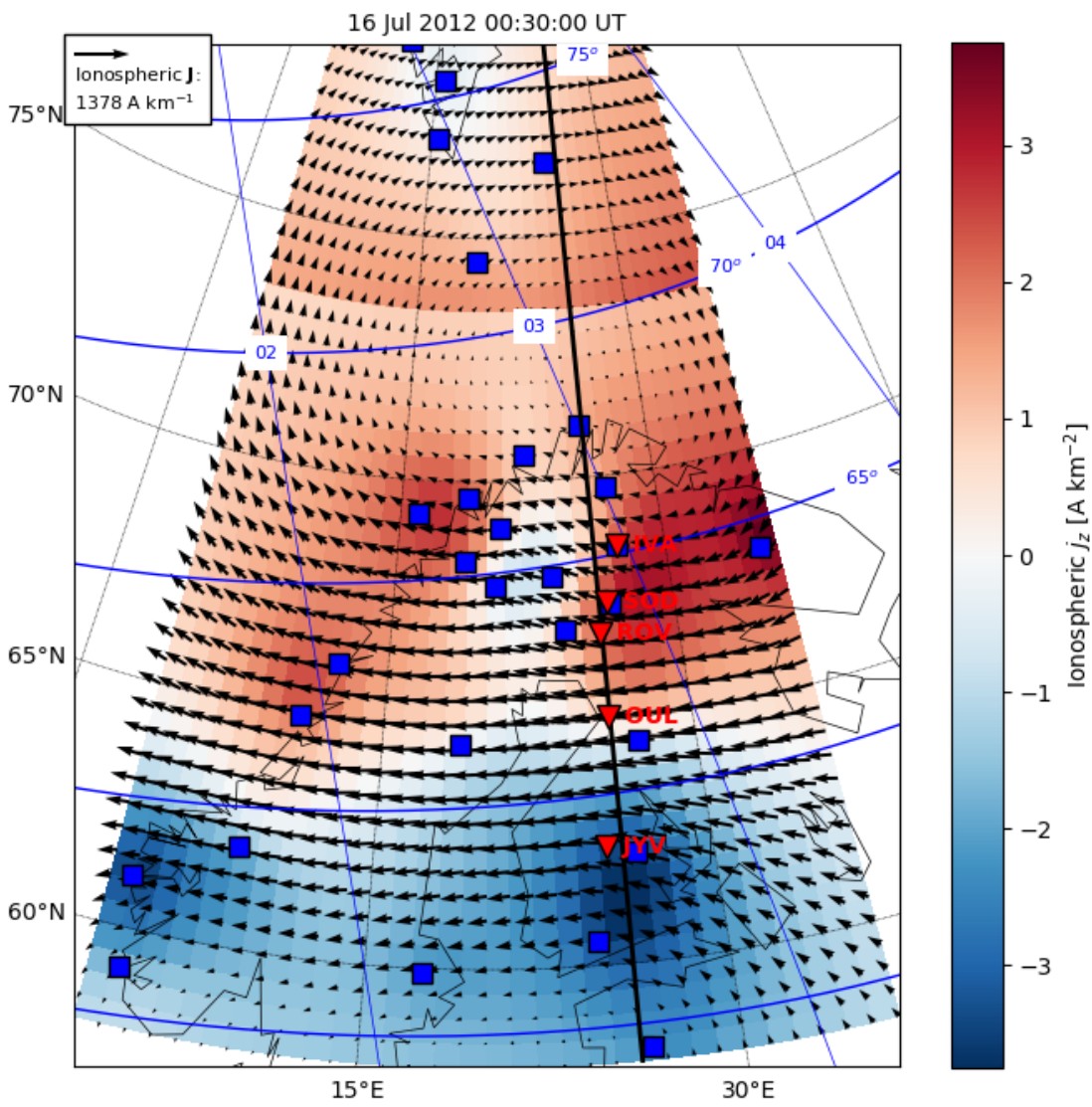

**Figure 1.** Locations of the riometer stations IVA, SOD, ROV, OUL, and JYV (red) and IMAGE magnetometers (blue). The vectors and colour show the ionospheric equivalent current density and its curl, derived from IMAGE data. The vertical black line marks the longitude along which latitude profiles of the east-west component of the equivalent current density and the curl have been extracted in order to create Figure 3. The blue grid indicates the magnetic latitudes and magnetic local times.





**Figure 2.** The ICME event of July 14–17, 2012. The top seven panels give a) the IMF magnitude, b) IMF components in GSM coordinates (blue: $B_X$, green: $B_Y$, red: $B_Z$) c) solar wind speed, d) solar wind dynamics pressure, e) subsolar magnetopause position from Shue et al. (1998), f) SYM–H (blue) and AL (grey). The bottom four panels (g-j) give riometer CNA data for IVA, SOD, OUL, and JYV. Time is given in MLT at the geomagnetic longitude of the riometer stations (here MLT ≈ UT + 2.5 hours). The vertical lines limit the sheath, flux rope and "back sheath" (see text for details).

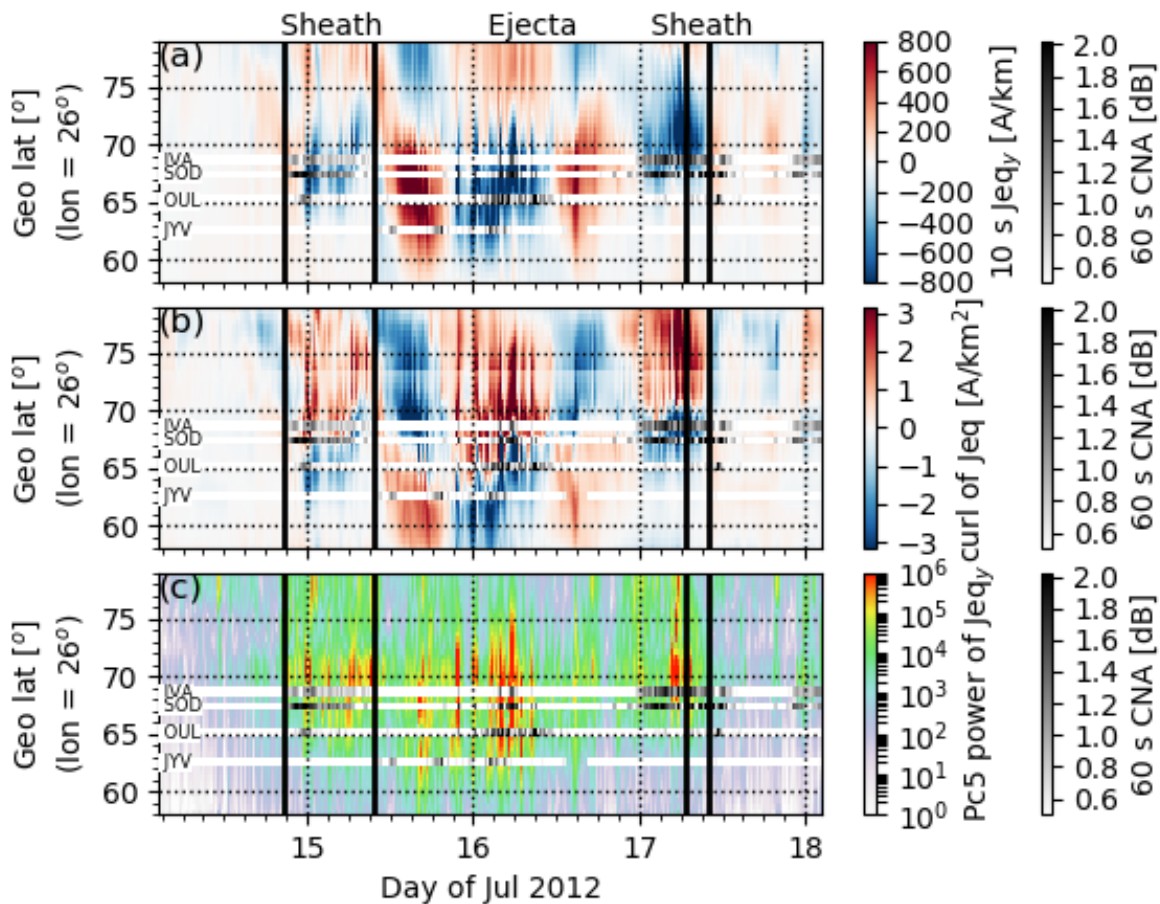

**Figure 3.** (a) Geographic latitude profiles of the auroral ionosphere eastward electrojet current density ($Jeq_y$) at 26° longitude, derived from 10 s IMAGE magnetometer measurements, (b) curl of the equivalent current density as a proxy for field-aligned current density (positive downward), and (c) Pc5 pulsation power, estimated by integrating $Jeq_y$ wavelet power over the Pc5 period range. All panels include 60 s cosmic noise absorption (CNA) as a function of UT, derived from IVA, SOD, OUL, and JYV riometer measurements. CNA can be used as an indicator of energetic (>30 keV) electron precipitation. Time is given in MLT at the geomagnetic longitude of the riometer stations (here MLT ≈ UT + 2.5 hours).

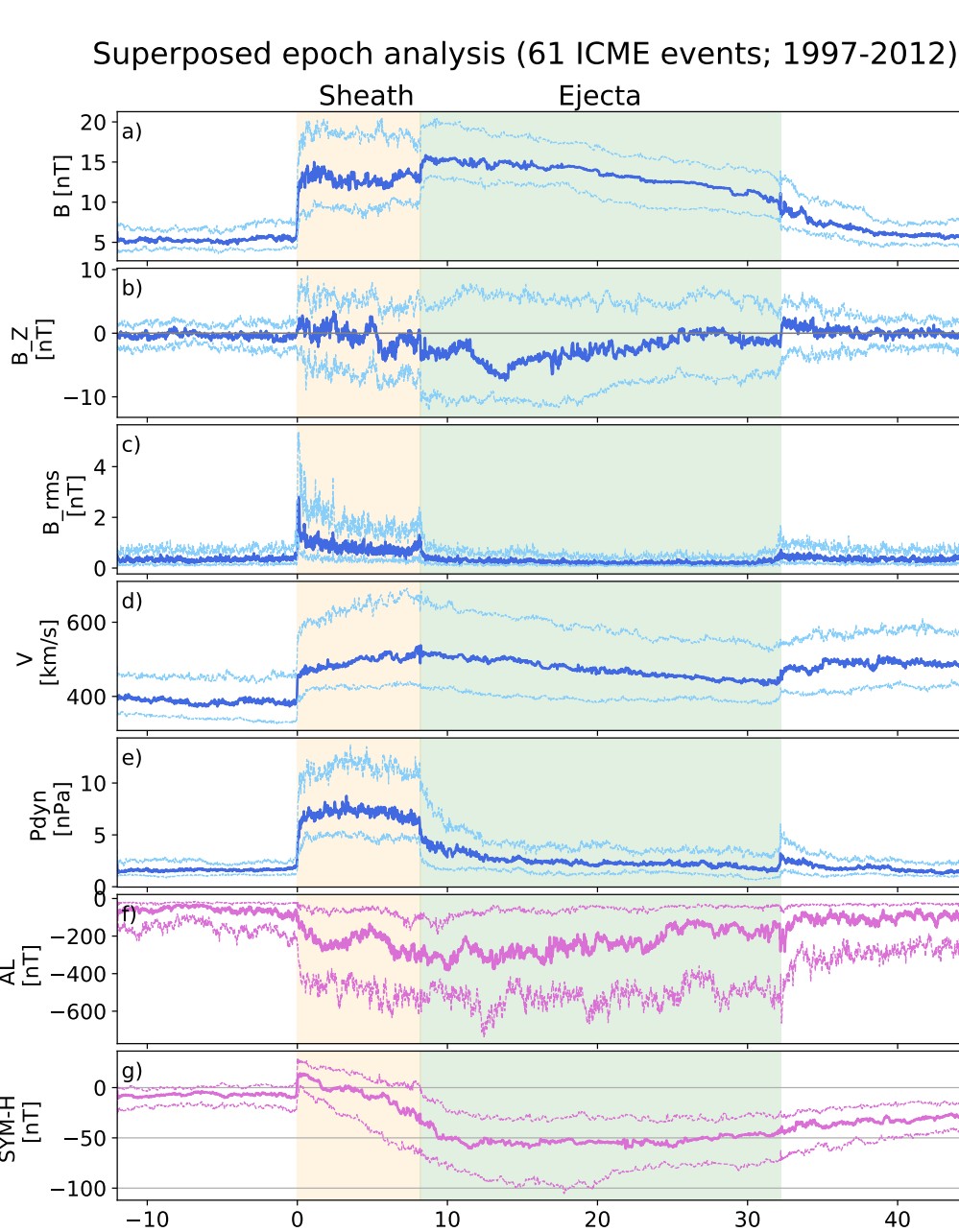

**Figure 4.** Superposed epoch analysis of solar wind conditions and geomagnetic indices during 61 ICME events detected in the near-Earth solar wind during 1997–2012. The thick curves show the median and thinner curves the lower and upper quartiles. The sheath and ejecta substructures are resampled to the same duration (see the text for details). The panels give: a) magnetic field magnitude, b) north-south magnetic field component, c) root-mean-square of the magnetic field, d) solar wind speed, e) solar wind dynamic pressure, f) AL index, g) SYMH index. The data is 1-minute OMNI data. Horizontal gray line in panel a) indicates $B_Z = 0$. In panel g) horizontal gray lines indicate SYMH$= 0$ nT, $-50$ nT (moderate storm limit) and $-100$ nT (intense storm limit).

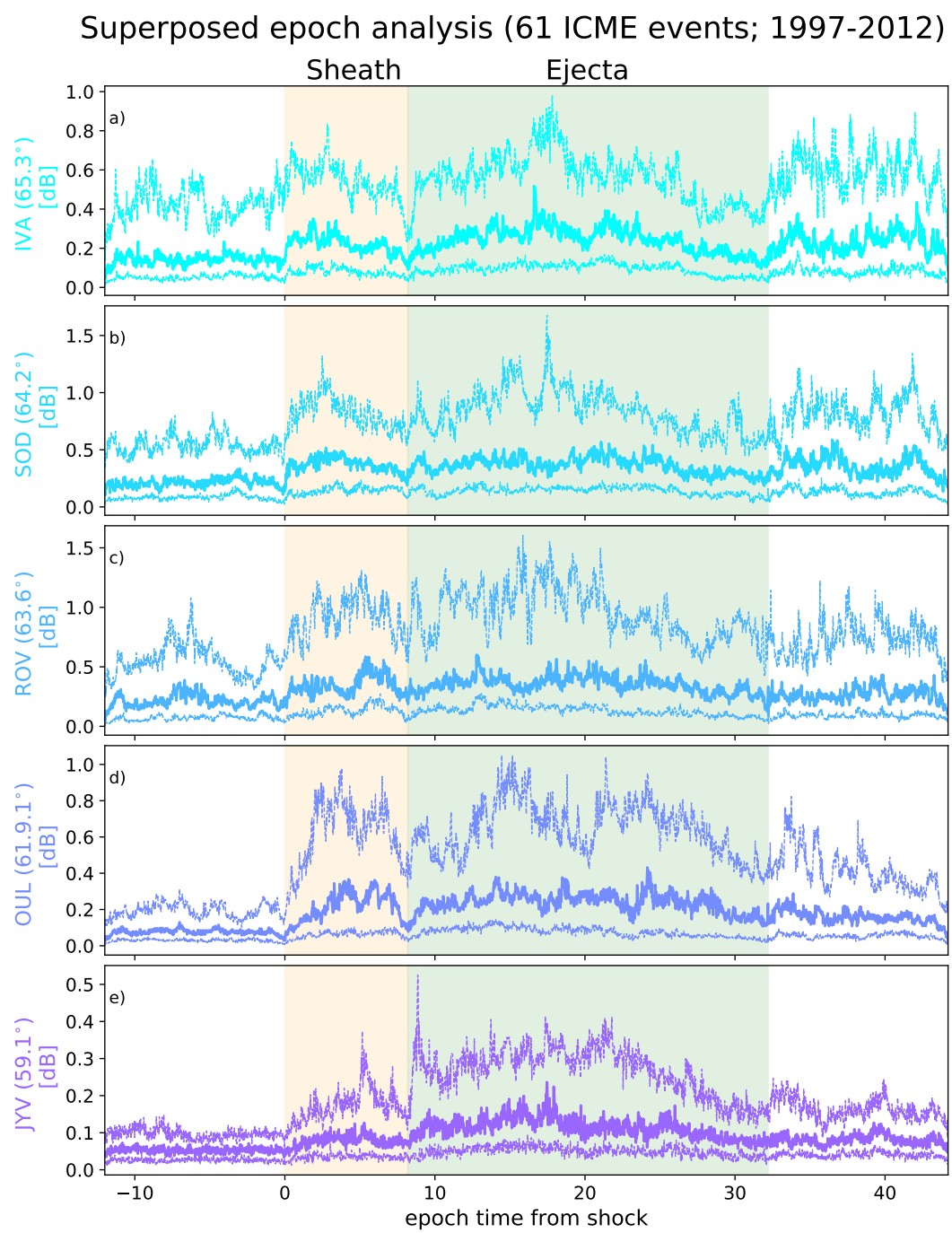

**Figure 5.** Superposed epoch analysis of CNA for 61 ICME events occurring between 1997–2012. The thick curves show the median and thinner curves the lower and upper quartiles. The data are shown for fice stations in the SGO chain (Ivalo, Sodankylä, Rovaniemi, Oulu and Jyväskylä) that are organized according to their MLAT from highest to lowest.

**Figure 6.** The relative occurrence frequency of CNA samples with $> 0.5$ dB for (top) sheath regions and (bottom) ejecta as a function of MLT (in 1 h bins). Stations are organized according to their MLAT that is indicated in parenthesis for each station.

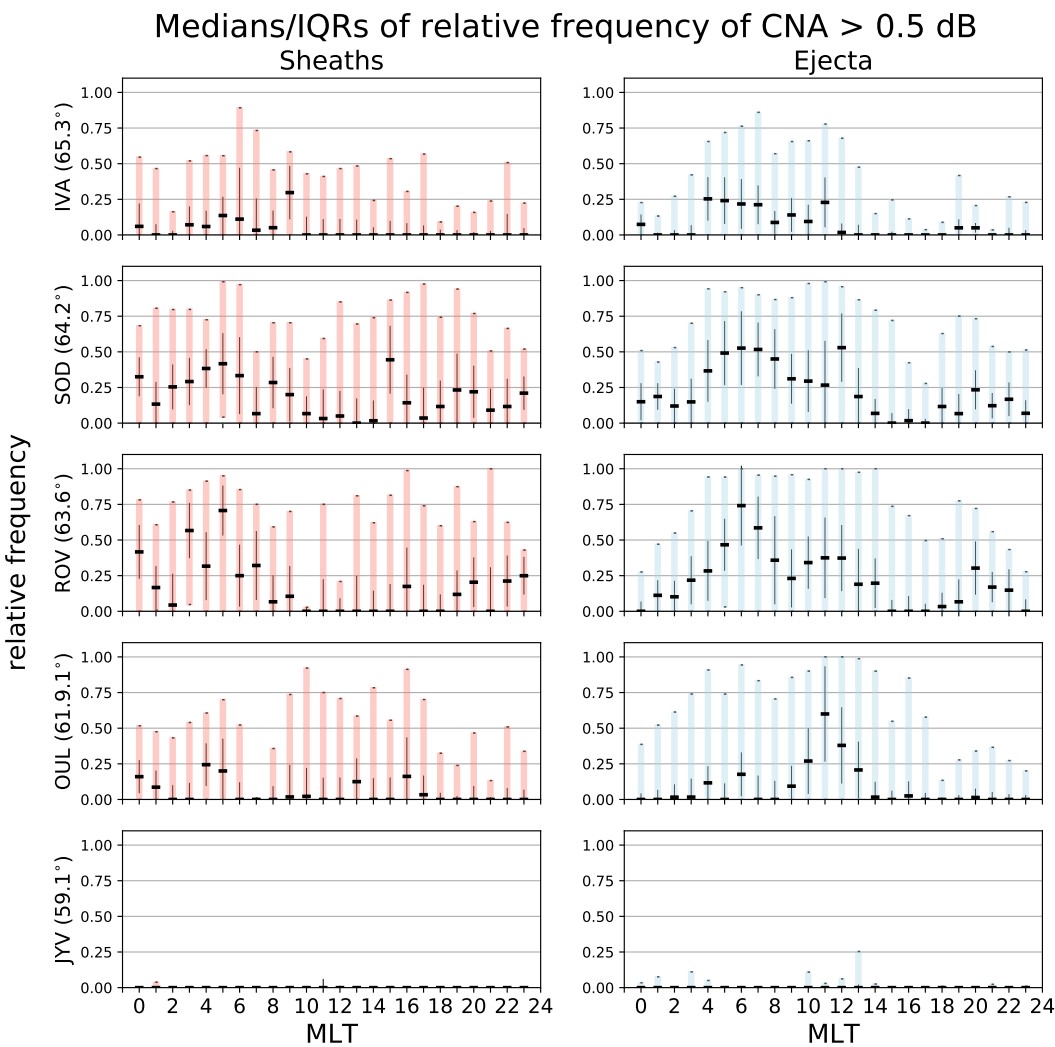

**Figure 7.** The medians and interquartial range (IQRs) calculated over relative occurrence frequencies of CNA > 0.5 are shown by the thick black horizontal bars and colored vertical bars dB for Left) sheaths and Right) ejecta. Stations are organized according to their MLAT that is indicated in parenthesis for each station. Vertical lines give the bootstrap errors calculated using 10 000 samples.

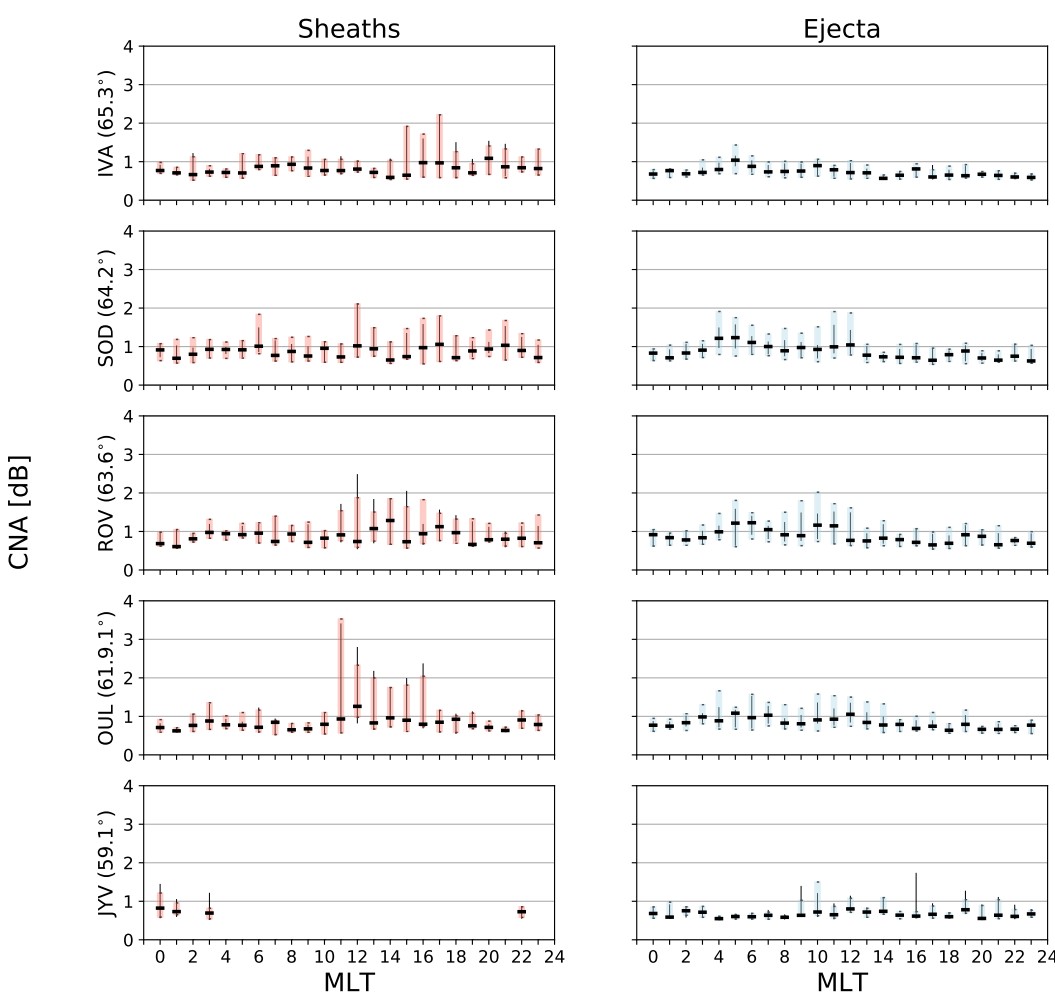

**Figure 8.** The medians (thick black horizontal bars) and interquartial ranges (IQRs) (colored boxes) of the CNA magnitude as a function of MLT for Left) sheaths and Right) ejecta. Only the cases with > 5 samples exceeding 0.5 dB during a 1-hour MLT bin are plotted. Stations are organized according to their magnetic latitude that is indicated in the parenthesis for each station. Vertical lines give the bootstrap errors calculated using 10 000 samples.



**Figure 9.** The medians (thick black horizontal bars) and interquartial range (IQR) (colored boxes) of Left) sheath and Right) ejecta properties and associated geomagnetic responses. The darker red and blue bars boxes show the IQRs for events associated with strongest CNA and lighter red and blue boxes the events associated with weakest CNA. The panels give a) interplanetary magnetic field magnitude, b) solar wind speed, c) dynamic pressure, d) magnetic field fluctuations estimated from root-mean-square, and e) AL and f) SYM–H indices.