# Peer review of "Cosmic noise absorption signature of particle precipitation during ICME sheaths and ejecta"

_Annales Geophysicae, 2019_

## Referee Comment (RC1) · Anonymous Referee #1 · 6 Jan 2020

Review of "Cosmic noise absorption signature of particle precipitation during ICME sheaths and ejecta" by Kilpua et al.

This paper examines the precipitation response during the two distinct phases of an ICME: the sheath and the ejecta. The rationale for this is the very different solar wind natures of the two components and the subsequent differences in the driving of the geomagnetic environment. The lead author has had past success in considering the different responses of geomagnetic parameters to the two components of the ICME. This study shows that there is a different >30 keV precipitation response during the two stages of the ICME; the authors use a chain of riometers in Finland and examine both the broad response in magnetic latitude and a more detailed variation in MLT. The authors provide a case study which I feel aids with the interpretation of the statistical

results and was a very good idea. The authors then perform an interesting version of the superposed epoch analysis whereby the 'sheath' and ejecta' for each events are normalized and resampled in time to make them all of equivalent average length. This interesting approach does show some differences in the two components, but masks a lot of the fine detail. The authors deal with this be delving into the MLT response, which sheds more light on the dominant drivers of the precipitation. All in all, I found this paper to be very interesting and is definitely worthy of eventual publication. It is well written, the figures are clear and well-chosen and it produces a new scientific result.

I have a number of comments that I feel need addressing, which I hope will help the authors improve the overall manuscript and some things for them to consider about their interpretation.

Comments: Page 1 Line 5: data as plural vs singular: 'The data come from' rather than 'comes from'

Page 2 Line 5: Hargreaves (and others) did much work on the morphology of CNA during substorm onset, with fine detail subsequently added later by the referenced authors. An outline including appropriate references appears in the Introduction to Kavanagh et al (2007), Ann. Geophys., 25, 1199-1214.

Page 2 Line 24: you may be interested in the paper by Kavanagh et al (2018), [JGR, 123(9). https://doi.org/10.1029/2018JA025890), who demonstrated significantly energetic precipitation within the plasmasphere following slot-region filling that lasted for days.

Page 2 Line 35 onwards: I am surprised you did not include Kavanagh et al. (2012) [JGR, 117 (A9), https://doi.org/10.1029/2011JA017320), in this list, given that they were the first to look at riometer data in the context of Co-rotating interaction regions in a statistical and superposed epoch analysis sense; using both an imaging riometer and a chain of riometers.

Page 3 Line 13: "Most of the studies consider ICMEs as a single entity,"

Page 4 Line 1: not 'used data sets', this makes it sound as if they are spent batteries. I suggest "we present the data used in the study".

Page 4 Line 2: "In Section 5 we discuss and summarize our results"

Page 4 Line 4: You ought to include the reference to the Little and Leinbach paper, although Shain predated them in the use of the technique it was Little and Leinbach that actually coined the term 'riometer'. Also the DOI associated with the Shain paper in the references cannot be found in the DOI System – there is a mistake somewhere.

Page 4 Line 7: why do you say 'so-called' with relation to CNA? This seems to suggest there is a problem with this name but you do not elaborate.

Page 4 Line 8: perhaps "height-integrated D-region electron density"

Table 1. I think from the text, the 'UT' header is actually the time to UT at MLT midnight. A note of caution – is this the same MLT calculation as used in Figure 1? It would be good to ensure you are consistent throughout. There is nothing wrong with a fixed MLT value applied to all UT, though it will become less accurate around dusk and dawn and will not cover the seasonal variation. When mixing data spread across several years (such as in a superposed epoch analysis), it does mean that the same 'bin' will have a mix of 'MLT' values within it. The bigger the bin, the less important this becomes, of course.

Page 5 Line 5: "for a review"

Page 5 Line 15: "corresponding to the "

Page 7 line 11: Did you do any spectral analysis on the riometer data? One of the key features of enhanced CNA during increased Pc5 power is that the CNA also shows a modulated signal. It would be interesting to know whether that happens in this case, and if not, speculate on why it does not, yet the CNA increases anyway.

Page 8 Line 5: I am having difficulty conceptualizing the 'double superposed epoch analysis' even after reading Kilpua et al 2015 (and then Kilpua 2013 [Ann Geo], which actually gives a slightly better explanation).

Given the importance of the analysis I think a little more description is called for. How are the data normalized in time? How is the resampling done? What effect does this have on the overall variability of the data set? What information is lost? What is gained?

If I understand the process correctly I think that this would diminish important features in the data. I suspect this is why the authors switch to the MLT response. Highlighting the method and its limitations might be useful for the les familiar reader.

Page 9 Line 7: "They presumably represent the population that had also the weakest geomagnetic response", presumably yes. I wonder if there is a way to test this? When determining the quartiles from the driver or even the geomagnetic index, is it feasible to track the corresponding CNA. I.e. for the values that make up the lowest quartiles in the AE, can you construct the median CNA from the corresponding values and contrast it with the lower quartile from the whole population?

Page 11 Line 18: something seems to have gone wrong with your '=' signs.

General Comments on the Discussion and figures 6 onwards

I do not necessarily disagree with your discussion and conclusions but wonder if the authors have missed some subtleties and potentially simpler explanations for the structures seen in the data. The thing that strikes me about the results displayed in these figures is how they match well with past studies. A frequency peak in the morning sector with a secondary peak in the evening (as evidenced for both sheath and ejecta in figure 7) is the standard pattern as first identified by Hargreaves (Planet Space Sci, 14,991-1006,1966 & with Cowley, Planet Space Sci, 1571-1583, 1967). This work was updated by Kavanagh et al (Ann Geophys., 22, 877-887, 2004).

This pattern shows some variation with activity and with magnetic latitude but as shown

in Kavanagh et al 2012 and Kavanagh et al 2007 (see above) it persists for major solar wind driving such as during co-rotating interaction regions and sawtooth events. I agree with the authors that it is driven by injection of plasma sheet electrons on the nightside that gradient-curvature drift eastward to dawn, interacting with chorus and precipitating (and energising at certain resonances and energies) through to the noon sector. During this sector the particles will be modulated by any ULF waves that happen to be present (either internally driven or externally).

In terms of the medians or magnitudes of CNA, rather than frequency of occurrence, I really like the result the authors have discovered. Figures 6 and 8 illustrate major differences in where the CNA is maximising in the sheaths relative to the ejecta. In the ejecta it is predominantly the familiar pattern. This makes sense to me as this is when the solar wind driving is likely to be more steady (e.g. see figure 4.) and the flux loading-unloading substorm cycle will dominate as the largest signal. On page 13, line 1 you suggest that the physical mechanism could be related to increased ULF Pc5 power driven by interplanetary shocks and pressure pulses. This is certainly a viable possibility. There is another possibility related to this, in that single shocks or pressure pulses can directly drive precipitation (see sudden impulse absorption) without the need for persistent wave activity. If a particle population is present, then a solar wind pressure pulse is highly likely to directly drive bursts of precipitation with relative delays in local time and magnetic latitude as the different propagation times of the compressional wave and the field-aligned wave. Much of the work began with:

Brown, R. R.: On the Poleward Expansion of Ionospheric Absorption Regions Triggered by Sudden Commencement of Geomagnetic Storms, J. Geophys. Res., 83, 1169–1171, 1978

Brown, R. R. and Driatsky, V. M.: Further Studies of Ionospheric and Geomagnetic Effects of Sudden Impulses, Planet. Space Sci.,21, 1931–1935, 1973

Brown, R. R., Hartz, T. R., Landmark, B., Leinbach, H., and Ortner, J.: Large-Scale

[Figure]

Electron Bombardment of the Atmosphere at the Sudden Commencement of a Geo-magnetic Storm, J. Geo-phys. Res., 66, 1035–1041, 1961

Perona, G. E.: Theory on the Precipitation of Magnetospheric Electrons at the Time of a Sudden Commencement, J. Geophys. Res.,77, 101–111, 1972

I wonder, might it be worth determining a PC5 power index for the riometer data for each event and looking at the timing and extent of that? Also what effect does a single pulse have on the index, by which I mean, does the PC5 index you used in the case study produce a strong value for an isolated amplitude increase (a spread delta function almost)? Or is that accounted for in the technique?

To summarise: I think this is a good paper that presents new and interesting results. Some of the interpretation of the results might benefit from increased familiarity with older work, particularly relating to sudden impulse/commencement absorption. I enjoyed the paper very much.

---

## Referee Comment (RC2) · Anonymous Referee #2 · 6 Jan 2020

This paper presents the statistical results of enhanced CNA corresponding to the sheath and ejecta structures of ICMEs using data from the Finnish riometer chain. The results showed that sheaths and ejecta are equally effective in inducing enhanced CNA. However, the occurrence frequency and the magnitude of enhanced CNA have different MLT distributions during sheaths and ejecta, which may reflect different MLT distribution of waves responsible for the energetic particle precipitation. The study is well-conceived and is appropriate for publication in this journal after considering the following comments.

Page 2 Line 6: "geostatationary" -> "geostationary"

Page 3 Line 20: "sheath" -> "sheaths" Please check all the "sheath" through the manuscript. "sheath" is a countable noun. So either use "sheathes" or "the/a sheath".

[Figure]

Page 3 Line 23: "at wide range of . . ." -> "at a wide range of. . ."

Page 3 Line 35: "substructres" -> "substructures"

Page 5 Line 11: "solar wind dynamics pressure" -> "solar wind dynamic pressure"

Page 8 Table 2: I would suggest using histograms instead to present the information presented in Table 2 so it would be easier for readers to get the latitudinal trend for each value.

Page 8 Line 5: It would be better to give a brief description of the "superposed epoch analysis" and explain more detailedly on how exactly you implement the method.

Page 22 Figure 2: "solar wind dynamics pressure" -> "solar wind dynamic pressure"

Page 25 Figure 5: "fice stations" -> "five stations"

---

## Author Comment (AC1) · 17 Jan 2020

**REFEREE #1**

This paper examines the precipitation response during the two distinct phases of an ICME: the sheath and the ejecta. The rationale for this is the very different solar wind natures of the two components and the subsequent differences in the driving of the geomagnetic environment. The lead author has had past success in considering the different responses of geomagnetic parameters to the two components of the ICME. This study shows that there is a different >30 keV precipitation response during the two stages of the ICME; the authors use a chain of riometers in Finland and examine both the broad response in magnetic latitude and a more detailed variation in MLT. The authors provide a case study which I feel aids with the interpretation of the statistical results and was a very good idea. The authors then perform an interesting version of the superposed epoch analysis whereby the 'sheath' and ejecta' for each events are normalized and resampled in time to make them all of equivalent average length. This interesting approach does show some differences in the two components, but masks a lot of the fine detail. The authors deal with this be delving into the MLT response, which sheds more light on the dominant drivers of the precipitation. All in all, I found this paper to be very interesting and is definitely worthy of eventual publication. It is well written, the figures are clear and well-chosen and it produces a new scientific result. I have a number of comments that I feel need addressing, which I hope will help the authors improve the overall manuscript and some things for them to consider about their interpretation.

**We thank the referee for his/her careful reading of our manuscript and many constructive comments and corrections. The suggestions were very helpful and have improved the manuscript. Please find below our detailed responses.**

Comments: Page 1 Line 5: data as plural vs singular: 'The data come from' rather than 'comes from'

**Corrected**

Page 2 Line 5: Hargreaves (and others) did much work on the morphology of CNA during substorm onset, with fine detail subsequently added later by the referenced authors. An outline including appropriate references appears in the Introduction to Kavanagh et al (2007), Ann. Geophys., 25, 1199-1214.

**Thanks for pointing these out. We have now added a few sentences describing in more detail precipitation signatures during different substorm phases and cited the papers mentioned above. We have also added Kavanagh et al., 2007 as a reference.**

Page 2 Line 24: you may be interested in the paper by Kavanagh et al (2018), [JGR, 123(9). https://doi.org/10.1029/2018JA025890), who demonstrated significantly energetic precipitation within the plasmasphere following slot-region filling that lasted for days.

**We have included this paper as a reference.**

Page 2 Line 35 onwards: I am surprised you did not include Kavanagh et al. (2012) [JGR, 117 (A9), https://doi.org/10.1029/2011JA017320), in this list, given that they were the first to look at riometer data in the context of Co-rotating interaction regions

in a statistical and superposed epoch analysis sense; using both an imaging riometer and a chain of riometers.

**Apologies for missing this paper, it is indeed highly relevant for our study. We have now included a discussion of this paper in the Introduction and Discussion sections.**

Page 3 Line 13: "Most of the studies consider ICMEs as a single entity,"

**corrected**

Page 4 Line 1: not 'used data sets', this makes it sound as if they are spent batteries. I suggest "we present the data used in the study".

**corrected**

Page 4 Line 2: "In Section 5 we discuss and summarize our results"

**corrected**

Page 4 Line 4: You ought to include the reference to the Little and Leinbach paper, although Shain predated them in the use of the technique it was Little and Leinbach that actually coined the term 'riometer'. Also the DOI associated with the Shain paper in the references cannot be found in the DOI System – there is a mistake somewhere.

**We have added Little and Leinbach as a reference. We checked the DOI of Shain et al. work and it seem to be wrong in ADS. We found the correct one (https://doi.org/10.1071/CH9510258) and have corrected it.**

Page 4 Line 7: why do you say 'so-called' with relation to CNA? This seems to suggest there is a problem with this name but you do not elaborate.

**We have removed "so-called"**

Page 4 Line 8: perhaps "height-integrated D-region electron density"

**Added**

Table 1. I think from the text, the 'UT' header is actually the time to UT at MLT midnight. A note of caution – is this the same MLT calculation as used in Figure 1? It would be good to ensure you are consistent throughout. There is nothing wrong with a fixed MLT value applied to all UT, though it will become less accurate around dusk and dawn and will not cover the seasonal variation. When mixing data spread across several years (such as in a superposed epoch analysis), it does mean that the same 'bin' will have a mix of 'MLT' values within it. The bigger the bin, the less important this becomes, of course.

**This was indeed sloppily marked in the table, it is the hours that are added to UT time to get approximate MLT time. We have now modified the table.**

Page 5 Line 5: "for a review"

**corrected**

Page 5 Line 15: "corresponding to the "

**corrected**

Page 7 line 11: Did you do any spectral analysis on the riometer data? One of the key features of enhanced CNA during increased Pc5 power is that the CNA also shows a modulated signal. It would be interesting to know whether that happens in this case, and if not, speculate on why it does not, yet the CNA increases anyway.

**We are not sure whether we have correctly understood this suggestion. We consider a deeper analysis of connection between riometer CNA signal and ULF Pc5 wave power a very important issue. We would like to however keep that as a separate study.**

Page 8 Line 5: I am having difficulty conceptualizing the 'double superposed epoch analysis' even after reading Kilpua et al 2015 (and then Kilpua 2013 [Ann Geo], which actually gives a slightly better explanation). Given the importance of the analysis I think a little more description is called for. How are the data normalized in time? How is the resampling done? What effect does this have on the overall variability of the data set? What information is lost? What is gained? If I understand the process correctly I think that this would diminish important features in the data. I suspect this is why the authors switch to the MLT response. Highlighting the method and its limitations might be useful for the les familiar reader.

**We have now extended this discussion significantly and added also links to a few other papers that have used a similar approach.**

Page 9 Line 7: "They presumably represent the population that had also the weakest geomagnetic response", presumably yes. I wonder if there is a way to test this? When determining the quartiles from the driver or even the geomagnetic index, is it feasible to track the corresponding CNA. I.e. for the values that make up the lowest quartiles in the AE, can you construct the median CNA from the corresponding values and contrast it with the lower quartile from the whole population?

**We removed this sentence since the medians etc. are calculated for each time-step from the set of events and geomagnetic response event wise is actually not straightforward to check in this context (i.e. it can vary from event to event as the time progresses).**

Page 11 Line 18: something seems to have gone wrong with your '=' signs.

**These should be approximate sign.**

General Comments on the Discussion and figures 6 onwards

I do not necessarily disagree with your discussion and conclusions but wonder if the authors have missed some subtleties and potentially simpler explanations for the structures seen in the data. The thing that strikes me about the results displayed in these figures is how they match well with past studies. A frequency peak in the morning sector with a secondary peak in the evening (as evidenced for both sheath and ejecta in figure 7) is the standard pattern as first identified by Hargreaves (Planet Space Sci, 14,991-1006,1966 & with Cowley, Planet Space Sci, 1571-1583, 1967). This work was

updated by Kavanagh et al (Ann Geophys., 22, 877-887, 2004).

This pattern shows some variation with activity and with magnetic latitude but as shown in Kavanagh et al 2012 and Kavanagh et al 2007 (see above) it persists for major solar wind driving such as during co-rotating interaction regions and sawtooth events. I agree with the authors that it is driven by injection of plasma sheet electrons on the nightside that gradient-curvature drift eastward to dawn, interacting with chorus and precipitating (and energising at certain resonances and energies) through to the noon sector. During this sector the particles will be modulated by any ULF waves that happen to be present (either internally driven or externally).

**We have added now in the beginning of this paragraph a discussion of Kavanagh et al. 2012 (reference to Hargreaves, 1966 also added) results and state that they are in agreement with our results, in particular considering the dawn/prenoon peak in the CNA signal. We also mention the fact that these have found to persist also for SIRs/fast streams and sawtooth events with references to Kavanagh et al. 2012 (+Grandin et al., 2017b) and Kavanagh et al. 2007. This is followed by discussion of the explanation being that substorm related electron drifts and subsequently scatter by chorus waves (where we now have added discussion of Kavanagh et al., 2012 results)**

In terms of the medians or magnitudes of CNA, rather than frequency of occurrence, I really like the result the authors have discovered. Figures 6 and 8 illustrate major differences in where the CNA is maximising in the sheaths relative to the ejecta. In the ejecta it is predominantly the familiar pattern. This makes sense to me as this is when the solar wind driving is likely to be more steady (e.g. see figure 4.) and the flux loading-unloading substorm cycle will dominate as the largest signal.

On page 13, line 1 you suggest that the physical mechanism could be related to increased ULF Pc5 power driven by interplanetary shocks and pressure pulses. This is certainly a viable possibility. There is another possibility related to this, in that single shocks or pressure pulses can directly drive precipitation (see sudden impulse absorption) without the need for persistent wave activity. If a particle population is present, then a solar wind pressure pulse is highly likely to directly drive bursts of precipitation with relative delays in local time and magnetic latitude as the different propagation times of the compressional wave and the field-aligned wave. Much of the work began with:

Brown, R. R.: On the Poleward Expansion of Ionospheric Absorption Regions Triggered by Sudden Commencement of Geomagnetic Storms, J. Geophys. Res., 83, 1169–1171, 1978

Brown, R. R. and Driatsky, V. M.: Further Studies of Ionospheric and Geomagnetic Effects of Sudden Impulses, Planet. Space Sci.,21, 1931–1935, 1973

Brown, R. R., Hartz, T. R., Landmark, B., Leinbach, H., and Ortner, J.: Large-Scale Electron Bombardment of the Atmosphere at the Sudden Commencement of a Geomagnetic Storm, J. Geo-phys. Res., 66, 1035–1041, 1961

Perona, G. E.: Theory on the Precipitation of Magnetospheric Electrons at the Time of a Sudden Commencement, J. Geophys. Res.,77, 101–111, 1972

**We have added a discussion of this in the Discussion and relevant references. This is indeed a good addition to the paper.**

I wonder, might it be worth determining a PC5 power index for the riometer data for each event and looking at the timing and extent of that? Also what effect does a single pulse have on the index, by which I mean, does the PC5 index you used in the case study produce a strong value for an isolated amplitude increase (a spread delta function almost)? Or is that accounted for in the technique?

**This is an excellent idea, but we would like to leave this for a more comprehensive future study.**

To summarise: I think this is a good paper that presents new and interesting results. Some of the interpretation of the results might benefit from increased familiarity with older work, particularly relating to sudden impulse/commencement absorption. I enjoyed the paper very much.

**We thank the referee for these kind words and are very happy to hear he/she finds our paper interesting. We have in particular now extended our discussion/interpretation and added a number of references to previous works.**

---

## Author Comment (AC2) · 17 Jan 2020

**REFEREE #2**

This paper presents the statistical results of enhanced CNA corresponding to the sheath and ejecta structures of ICMEs using data from the Finnish riometer chain. The results showed that sheaths and ejecta are equally effective in inducing enhanced CNA. However, the occurrence frequency and the magnitude of enhanced CNA have different MLT distributions during sheaths and ejecta, which may reflect different MLT distribution of waves responsible for the energetic particle precipitation. The study is well-conceived and is appropriate for publication in this journal after considering the following comments.

**We thank the referee for his/her careful reading of our manuscript and constructive comments and corrections. We have modified the paper accordingly.**

Page 2 Line 6: "geostatationary" -> "geostationary"

**Corrected**

Page 3 Line 20: "sheath" -> "sheaths" Please check all the "sheath" through the manuscript. "sheath" is a countable noun. So either use "sheathes" or "the/a sheath"

**We have checked through the manuscript and corrected from several places sheath → sheaths**

Page 3 Line 23: "at wide range of : : :" -> "at a wide range of: : :"

**Corrected**

Page 3 Line 35: "substructres" -> "substructures"

**Corrected**

Page 5 Line 11: "solar wind dynamics pressure" -> "solar wind dynamic pressure"

**Corrected**

Page 8 Table 2: I would suggest using histograms instead to present the information presented in Table 2 so it would be easier for readers to get the latitudinal trend for each value.

**We would like to keep this information in a table format, partly also because we have quite many figures already in the paper.**

Page 8 Line 5: It would be better to give a brief description of the "superposed epoch analysis" and explain more detailedly on how exactly you implement the method.

**We have added a more detailed description.**

Page 22 Figure 2: "solar wind dynamics pressure" -> "solar wind dynamic pressure"

**Corrected**

Page 25 Figure 5: "fice stations" -> "five stations"

**Corrected**